# SignRFF: Sign Random Fourier Features

**Xiaoyun Li,   Ping Li**

LinkedIn Ads
700 Bellevue WA NE, Bellevue, WA 98004, USA
{lixiaoyun996,pingli98}@gmail.com

## Abstract

The industry practice has been moving to embedding based retrieval (EBR). For example, in many applications, the embedding vectors are trained by some form of two-tower models. During serving phase, candidates (embedding vectors) are retrieved according to the rankings of cosine similarities either exhaustively or by approximate near neighbor (ANN) search algorithms. For those applications, it is natural to apply "sign random projections" (SignRP) or variants, on the trained embedding vectors to facilitate efficient data storage and cosine distance computations. SignRP is also one of the standard indexing schemes for conducting approximate near neighbor search. In the literature, SignRP has been popular and, to an extent, becomes the default method for "locality sensitive hashing" (LSH).

In this paper, we propose "sign random Fourier features" (SignRFF) as an alternative to SignRP. The original method of random Fourier features (RFF) is a standard technique for approximating the Gaussian kernel (as opposed to the linear cosine kernel), in the literature of large-scale machine learning. Basically, RFF applies a simple nonlinear transformation on the samples generated by random projections (RP). Thus, in the pipeline of EBR, it is straightforward to replace SignRP by SignRFF. This paper explains, in a principled manner, why it makes sense to do so.

In this paper, a new analytical measure called **Ranking Efficiency (RE)** is developed, which in retrospect is closely related to the "two-sample mean" $t$-test statistic for binomial variables. RE provides a systematic and unified framework for comparing different LSH methods. We compare our proposed SignRP with SignRP, KLSH (kernel LSH), as well SQ-RFF (which is another 1-bit coding scheme for RFF). According to the RE expression, SignRFF consistently outperforms KLSH (for Gaussian kernel) and SQ-RFF. SignRFF also outperforms SignRP in the relatively high similarity region. The theoretical comparison results are consistent with our empirical findings. In addition, experiments are conducted to compare SignRFF with a wide range of data-dependent and deep learning based hashing methods and show the advantage of SignRFF with a sufficient number of hash bits.

## 1  Introduction

We study the search problem with data points in $\mathbb{R}^d$. Let $\boldsymbol{X}$ denote the database consisting of $n$ data points. Given a query data point $\boldsymbol{q}$, the task is to find the a similar data point in $\boldsymbol{X}$. For the purpose of discussion, we denote two data points in $\boldsymbol{X}$ by $\boldsymbol{x}$ and $\boldsymbol{y}$. For simplicity, we consider the data points are normalized, i.e., $\|\boldsymbol{x}\| = \|\boldsymbol{y}\| = 1$. Therefore, the "cosine" similarity between $\boldsymbol{x}$ and $\boldsymbol{y}$ is simply $\rho = \cos(\boldsymbol{x}, \boldsymbol{y}) = \sum_{i=1}^{d} x_i y_i$. We also denote the Gaussian kernel between $\boldsymbol{x}$ and $\boldsymbol{y}$ to be $k(\rho) \equiv k(\boldsymbol{x}, \boldsymbol{y}) = e^{-\gamma^2(1-\rho)}$, where $\gamma > 0$ is a tuning parameter. In this paper, we study binary coding algorithms for efficiently finding near neighbors based on similarities related to $\rho$ and $k(\rho)$. In particular, our methods are based on random projections and (quantized) random Fourier features.

36th Conference on Neural Information Processing Systems (NeurIPS 2022).

## 1.1 Random Projections (RP) and Sign Random Projections (SignRP)

Denote by $\boldsymbol{w}$ a $d$-dim vector of random i.i.d. entries, i.e., $w_i \sim N(0,1)$, $i = 1$ to $d$. The idea of random projections (RP) is to compute the inner product $\boldsymbol{w}^T \boldsymbol{x}$ between $\boldsymbol{w}$ and the data vector $\boldsymbol{x}$, and the same for $\boldsymbol{w}^T \boldsymbol{y}$. It is easy to see that $\mathbb{E}[(\boldsymbol{w}^T \boldsymbol{x})(\boldsymbol{w}^T \boldsymbol{y})] = \cos(\boldsymbol{x}, \boldsymbol{y}) = \rho$. The idea of sign random projections (SignRP) is to keep only the signs of the projected values:

$$\textbf{SignRP:} \quad h_{RP}(\boldsymbol{x}) = sign(\boldsymbol{w}^T \boldsymbol{x}), \quad h_{RP}(\boldsymbol{y}) = sign(\boldsymbol{w}^T \boldsymbol{y}). \tag{1}$$

There is a chance that the two hash values, i.e., the signs $h_{RP}(\boldsymbol{x})$ and $h_{RP}(\boldsymbol{y})$, will be equal [16, 5]:

$$P(h_{RP}(\boldsymbol{x}) = h_{RP}(\boldsymbol{y})) = 1 - \frac{\cos^{-1}(\rho)}{\pi}. \tag{2}$$

Interestingly, if $w_i$ is sampled from a Cauchy distribution (instead of Gaussian), then the collision probability is closely related to the popular Chi-square similarity instead of the cosine [37].

## 1.2 Binary Code and Approximate Near Neighbor (ANN) Search

The method of sign random projections (SignRP) and the collision probability (2) have been widely used [5] as a standard hashing method for locality sensitive hashing (LSH) in the context of approximate near neighbor (ANN) search [21]. Basically, we can generate $b$ independent hashes to form a hash table of size $2^b$ (e.g., $b = 20$ means a hash table with 1 million buckets). For a query $\boldsymbol{q}$, one can compute $b$ hash values $h_{RP}(\boldsymbol{q})$ to form a $b$-bit code and retrieve all data points form $\boldsymbol{X}$ which fall into the bucket corresponding to the $b$-bit code. To improve the retrieval accuracy, typically multiple hash tables are used to return the union of retrieved results from all the tables.

SignRP is one example of a broad class of method based on binary coding for ANN. That is, for each data vector $\boldsymbol{x} \in \mathbb{R}^d$, we hash it into a length-$b$ binary 0/1 vector $h(\boldsymbol{x}) \in \{0, 1\}^b$, where the geometry of the data should be well preserved in the hamming space. Searching with binary codes has been widely applied in many applications, such as large-scale image retrieval [65, 17, 26, 20, 44, 45, 46, 61, 72, 68]. Besides the benefit of storage saving (especially for high-dimensional data), compressing data to binary can also significantly speedup the retrieval process. In this paper, our analysis and evaluation will focus on the hamming ranking approach which has been widely applied in image/video retrieval [26, 17, 49, 50, 4, 56, 70, 18, 47, 52], where we conduct exhaustive search in the hamming space. In fact, SignRP is also used by other ANN systems such as graph-based ANN as a crucial component to save the storage and speedup distance computations [76, 75].

## 1.3 Random Fourier Features (RFF) and SignRFF (Our Contribution)

Nonlinear kernels have been proven to be effective in many machine learning tasks [57, 2, 19]. See [33] for a study which empirically compared specially designed nonlinear kernels (i.e., "tunable GMM kernels") with deep nets and boosted trees. In this study, we focus on the Gaussian kernel:

$$k(\rho) \equiv k(\boldsymbol{x}, \boldsymbol{y}) = e^{-\gamma^2(1-\rho)}.$$

One can also utilize random projections to approximate the Gaussian kernel via the random Fourier features (RFF) [55, 54]. For data vectors $\boldsymbol{x}, \boldsymbol{y} \in \mathbb{R}^d$, the RFF for the Gaussian kernel is defined as

$$\textbf{RFF:} \quad F(\boldsymbol{x}) = \cos(\boldsymbol{w}^T \boldsymbol{x} + \tau), \quad F(\boldsymbol{y}) = \cos(\boldsymbol{w}^T \boldsymbol{y} + \tau), \tag{3}$$

where $\boldsymbol{w} \sim N(0, \gamma^2 \boldsymbol{I}_d)$ and $\tau \sim Unif(0, 2\pi)$ where $\boldsymbol{I}_d$ is the identity matrix. It holds that $\mathbb{E}[F(\boldsymbol{x})F(\boldsymbol{y})] = k(\boldsymbol{x}, \boldsymbol{y})/2$, i.e., the non-linear kernel can be preserved in expectation by the linear inner product of RFF. To approximate the kernel, we generate $b$ independent RFFs for each data point using i.i.d. $\boldsymbol{w_1}, ..., \boldsymbol{w_b}$ and $\tau_1, ..., \tau_b$, which can be used for subsequent learning tasks. Additionally, it was shown in [32] that the normalized RFFs (NRFF) can substantially reduce the variance of RFF.

This method has lead to many applications in large-scale learning where one trains linear models on RFF to approximate non-linear kernel machines [71, 9, 63, 1, 62, 64, 39]. To extend RFF to retrieval tasks, in this paper, we propose SignRFF by keeping only the signs of the random Fourier features:

$$\textbf{SignRFF:} \quad h_{sign}(\boldsymbol{x}) = sign(F(x)) = sign(\cos(\boldsymbol{w}^T \boldsymbol{x} + \tau)). \tag{4}$$

We skip the same step for $h_{sign}(y)$. Note that, this is different from the earlier work called "SQ-RFF" [53] which proposed to construct binary codes from RFFs using stochastic binary quantization:

$$\textbf{SQ-RFF:} \quad h_{SQ}(\boldsymbol{x}) = sign(F(\boldsymbol{x}) + \xi) = sign(\cos(\boldsymbol{w}^T \boldsymbol{x} + \tau) + \xi), \tag{5}$$

where $\xi \sim Unif(-1, 1)$ is a random perturbation. To an extent, our idea was inspired by [36], which showed that the stochastic perturbation in [11] for quantized random projections was not needed. In this study, we will show theoretically and empirically that SignRFF is better than SQ-RFF. In particular, we propose a new evaluation metric named "ranking efficiency" (RE) to serve as a unified measure to analytically compare the search performance of different LSH methods. Under this metric, SignRFF consistently outperforms various other LSH schemes. Our experimental study also confirms that RE is a strong predictor of the empirical search performances.

### 1.4 Other Related Methods

Quantization methods for random projections have been extensively studied in the literature, e.g., [16, 5, 11, 13, 36, 34, 41, 40]. Quantized methods for random Fourier features have also been heavily-studied [53, 74, 43, 43, 42]. In the meanwhile, there have been many works which have focused on learning *data-dependent* binary hash codes, through different objective functions. Examples include Iterative Quantization (ITQ) [17], Spectral Hashing (SpecH) [65] and Binary Reconstruction Embedding (BRE) [26]. Recently, some unsupervised deep learning based methods have been proposed, many of which are, to some extent, "task-specific" for cross-modal/video/image retrieval, implemented based on some deep models like the autoencoder and VGG-16 [45, 12, 8, 38, 69, 18, 47, 52], showing promising performance in image retrieval tasks by taking advantage of the complicated model structures (e.g., CNN layers) [69, 52].

## 2 Background: Locality-Sensitive Hashing (LSH)

As mentioned earlier, in large-scale information retrieval, exhaustive search of the exact nearest neighbors is usually too expensive. A common relaxation in this setting is the Approximate Nearest Neighbor (ANN) search [21], where we return a "good" neighborhood of a query with high probability. In this paper, we consider the search problem with data points in $\mathbb{R}^d$. $\boldsymbol{X}$ denotes the database consisting of $n$ data points, and $\boldsymbol{q}$ is a query point. Recall that $\boldsymbol{x}, \boldsymbol{y}$ are two data points with $\rho = \cos(\boldsymbol{x}, \boldsymbol{y})$.

**Definition 1** ($\tilde{S}$-neighbor). *For a similarity measure $S : \mathbb{R}^d \times \mathbb{R}^d \mapsto \mathbb{R}$, the $\tilde{S}$-neighbor set of $\boldsymbol{q}$ is defined as $\{\boldsymbol{x} \in \boldsymbol{X} : S(\boldsymbol{x}, \boldsymbol{q}) > \tilde{S}\}$.*

**Definition 2** ($(c, \tilde{S})$-ANN). *An algorithm $\mathbb{A}$ is a $(c, \tilde{S})$-ANN method provided the following: with probability at least $1 - \delta$, for $0 < c < 1$, if there exists an $\tilde{S}$-neighbor of $\boldsymbol{q}$ in $\boldsymbol{X}$, $\mathbb{A}$ returns a $c\tilde{S}$-neighbor of $\boldsymbol{q}$, where $\delta > 0$ is a parameter.*

One popular family of hash functions satisfying Definition 2 is the Locality-Sensitive Hashing (LSH), whose general definition is provided below.

**Definition 3** ([21]). *A family of hash functions $\mathcal{H}$ is called $(\tilde{S}, c\tilde{S}, p_1, p_2)$-locality-sensitive for similarity measure $S$ and $0 < c < 1$, if for $\forall \boldsymbol{x}, \boldsymbol{y} \in \mathbb{R}^d$ and hash function $h$ uniformly chosen from $\mathcal{H}$, it holds that: 1) If $S(\boldsymbol{x}, \boldsymbol{y}) \geq \tilde{S}$, then $P(h(\boldsymbol{x}) = h(\boldsymbol{y})) \geq p_1$; 2) If $S(\boldsymbol{x}, \boldsymbol{y}) \leq c\tilde{S}$, then $P(h(\boldsymbol{x}) = h(\boldsymbol{y})) \leq p_2$, with $p_2 < p_1$.*

A key intuition of LSH is that, similar data points will have a higher chance of hash collision in the Hamming space. One example of LSH method, associated with the cosine similarity, is given by the SignRP approach as in (1). Note that the hash collision probability (2) is increasing in $\rho$, which, by Definition 3, is the key to ensure the locality sensitivity of SignRP.

Compared with the data-dependent methods (e.g., as mentioned in Section 1.4), LSH has several advantages. Firstly, although data-dependent procedures can provide improved performance with fewer binary codes, a known weakness of many of these mechanisms is the performance bottleneck when we increase the code length $b$ [53, 24]. On the other hand, the search performance of the data-independent LSH would keep boosting with larger $b$. In many scenarios where short codes (e.g., $\leq 128$ bits) cannot achieve a desirable level of search accuracy for practical purposes, using

longer LSH codes could be more favorable. Moreover, LSH is very simple to implement (only with random projections), while data-dependent methods require additional optimization/training and longer inference time (e.g., for deep networks). Lastly, LSH enjoys rigorous theoretical guarantees on the retrieval performance. Therefore, LSH has been a popular hashing method with many practical applications for decades [3, 5, 11, 58, 59, 31, 51, 14, 7, 35, 73, 28, 30, 10, 6, 66, 67].

**Kernelized LSH (KLSH).** The SignRP (1) approach is based on linear random projections. Recall the Gaussian kernel function defined for $\boldsymbol{x}, \boldsymbol{y} \in \mathbb{R}^d$ as $k(\rho) \equiv k(\boldsymbol{x}, \boldsymbol{y}) = e^{-\gamma^2(1-\rho)}$, where $\gamma$ is a hyper-parameter. Let $\Psi : \mathbb{R}^d \mapsto \mathcal{F}$ be the feature map to the kernel induced feature space $\mathcal{F}$. To incorporate non-linearity in the hash codes, [27] proposed Kernelized Locality-Sensitive Hashing (KLSH) by conceptually applying SignRP (1) in the kernel induced feature space $\mathcal{F}$, i.e., $h(\boldsymbol{x}) = sign(\boldsymbol{w}^T \Psi(\boldsymbol{x}))$. As in many cases (e.g., for the Gaussian kernel) the map $\Psi$ cannot be explicitly identified, KLSH proposes to approximate the random Gaussian distribution using data through the Central Limit Theorem (CLT) in the Reproducing Kernel Hilbert Space. Specifically, we first sample $m$ data points from $\boldsymbol{X}$ to form a kernel matrix $\boldsymbol{K}$, then uniformly pick $t$ points from $[1, ..., m]$ at random to approximate the Gaussian distribution. After some algebra, the hash code for $\boldsymbol{x} \in \mathbb{R}^d$ is finally computed as

$$\textbf{KLSH:} \quad h_{KLSH}(\boldsymbol{x}) = sign(\sum_{i=1}^{m} \boldsymbol{r}(i) k(\boldsymbol{x}, \boldsymbol{x}_i)), \tag{6}$$

where $\boldsymbol{r} = \boldsymbol{K}^{-1/2} \boldsymbol{e}_t$, and $\boldsymbol{e}_t \in \{0, 1\}^m$ has ones at the indices of the $t$ selected points. Since KLSH uses a pool of data samples to approximate the Gaussian distribution, the hash codes are in fact dependent in implementation. Thus, a performance bottleneck has also been observed for KLSH as $b$ increases [24], similar to many data-dependent methods. See Appendix D for more explanation.

**Embedding based retrieval (EBR)** has become an mainstream in industry practice, owing to the matured technologies in deep representation learning and approximate near neighbor (ANN) search. For example, [15] described how to use a two-tower model to train embedding vectors using click-through data, and then use ANN techniques in the serving phase to retrieve ads candidates based on the cosine similarities of embedding vectors. [15] demonstrated the advantages of EBR in terms of improvements in CTR (click-through rate) and ads revenue. As EBR has become the standard practice in industry, a universal problem arises, that is, how to effectively store the embeddings and efficiently compute cosine similarities. SignRP and variants such as [13, 36, 34, 41] would be an option for EBR for multiple purposes: data reduction, efficient distance computation, as well as indexing for ANN. On the other hand, we can use hashing schemes (SignRP, KLSH, SQ-RFF, etc.) which are related to the Gaussian kernel, to replace SignRP. Of course, one can also choose hashing methods for similarities which are not cosine or Gaussian. One such example is the generalize min-max kernel and consistent weighted sampling (and their variants) [48, 22, 32, 35].

## 3  Locality-Sensitive Hashing From Random Fourier Features

Random Fourier feature (RFF) [55, 54] is a tool for alleviating the computational bottleneck of standard kernel methods. For a data vector $\boldsymbol{x} \in \mathbb{R}^d$, recall (3) the RFF for the Gaussian kernel as

$$\textbf{RFF:} \quad F(\boldsymbol{x}) = \cos(\boldsymbol{w}^T \boldsymbol{x} + \tau),$$

where $\boldsymbol{w} \sim N(0, \gamma^2 \boldsymbol{I}_d)$ and $\tau \sim Unif(0, 2\pi)$ where $\boldsymbol{I}_d$ is the identity matrix. It holds that $\mathbb{E}[F(\boldsymbol{x})F(\boldsymbol{y})] = k(\boldsymbol{x}, \boldsymbol{y})/2$. The probability distribution of RFF is given as follows.

**Lemma 1** ([43])**.** *For two normalized data points $\boldsymbol{x}, \boldsymbol{y}$ with cosine $\rho$, let $F(\cdot)$ be the RFF defined as (3). The joint distribution of $z_x = F(x)$ and $z_y = F(y)$ is*

$$f(z_x, z_y | \rho) = \frac{\sum\limits_{k=-\infty}^{\infty} \left[ \phi_\sigma(a_x^* - a_y^* + 2k\pi) + \phi_\sigma(a_x^* + a_y^* + 2k\pi) \right]}{\pi \sqrt{1 - z_x^2} \sqrt{1 - z_y^2}},$$

*where $a_x^* = \cos^{-1}(z_x), a_y^* = \cos^{-1}(z_y)$, and $\phi_\sigma(\cdot)$ is the p.d.f. of $N(0, \sigma^2)$ with $\sigma = \sqrt{2(1-\rho)}\gamma$. Furthermore, $\mathbb{E}[sign(F(\boldsymbol{x})) sign(F(\boldsymbol{y}))]$ is an increasing function of $\rho$.*

## 3.1 SQ-RFF with Stochastic Quantization

To extend RFF to efficient search algorithms, [53] designed a mapping $[-1, 1] \mapsto \{0, 1\}$ to construct binary codes from RFF. For $\boldsymbol{x} \in \mathbb{R}^d$, the code is produced by stochastic quantization (SQ):

$$\textbf{SQ-RFF:} \quad h_{SQ}(\boldsymbol{x}) = sign(F(\boldsymbol{x}) + \xi) = sign(\cos(\boldsymbol{w}^T\boldsymbol{x} + \tau) + \xi),$$

where $\xi$ is a random perturbation from $Unif(-1, 1)$. Effectively, the so-called SQ-RFF applies a sampling procedure where we first compute the RFF $z = \cos(\boldsymbol{w}^T\boldsymbol{x} + \tau)$, and then assign it to 1 (otherwise $-1$) with probability $\frac{|1+z|}{2}$. This approach has been used in [74, 43] for large-scale low-precision kernel training. The collision probability of "SQ-RFF" is given below.

**Theorem 1** ([53]). *For SQ-RFF (5), for normalized $\boldsymbol{x}, \boldsymbol{y} \in \mathbb{R}^d$ with $\cos(\boldsymbol{x}, \boldsymbol{y}) = \rho$, it holds that*

$$P_{SQ}(\rho) := P(h_{SQ}(\boldsymbol{x}) = h_{SQ}(\boldsymbol{y})) = 1 - \frac{8}{\pi^2} \sum_{s=1}^{\infty} \frac{1 - e^{-\gamma^2 s^2 (1-\rho)}}{4s^2 - 1}. \tag{7}$$

**Proposition 1.** *The SQ-RFF in (5) is $(\tilde{k}, c\tilde{k}, P_{SQ}(\rho_1), P_{SQ}(\rho_2))$-locality sensitive w.r.t. similarity measure $k(\cdot)$, where $\rho_1 = \log(\tilde{k})/\gamma^2 + 1$ and $\rho_2 = \log(c\tilde{k})/\gamma^2 + 1$.*

*Proof.* According to Definition 3, SQ-RFF is locality-sensitive w.r.t. $\rho$ because (7) is an increasing function of $\rho$. Hence, it is also locality-sensitive w.r.t. the kernel $k(\rho)$ by the monotonicity of $k(\rho)$. The $\rho_1$ and $\rho_2$ are derived by inserting $\tilde{k}$ into the inverse map of the kernel function. $\qquad\square$

## 3.2 SignRFF: Why Not Drop the Noise?

The SQ-RFF has been a standard approach for constructing binary codes from RFF for over a decade. Yet, is the additional perturbation added to the RFF before binarization really necessary? Motivated by this question, we propose a simpler coding strategy to directly take the sign of RFF (i.e., deterministic quantization) and remove the noise $\xi$. Formally, the SignRFF approach is defined by

$$\textbf{SignRFF:} \quad h_{sign}(\boldsymbol{x}) = sign(F(x)) = sign(\cos(\boldsymbol{w}^T\boldsymbol{x} + \tau)).$$

Operationally, SignRFF is extremely convenient. At the first glance, it may appear a bit surprising that this simple scheme has not been studied in literature. We believe one of the reasons might be that the theoretical correctness of SignRFF was hard to justify. Based on the joint density function of RFF, we show that SignRFF indeed belongs to the LSH family.

**Proposition 2.** *The SignRFF (4) is $(\tilde{k}, c\tilde{k}, P_{sign}(\rho_1), P_{sign}(\rho_2))$-locality sensitive w.r.t. to $k(\cdot)$, with $\rho_1 = \log(\tilde{k})/\gamma^2 + 1$ and $\rho_2 = \log(c\tilde{k})/\gamma^2 + 1$, with collision probability*

$$P_{sign}(\rho) := P(h_{sign}(\boldsymbol{x}) = h_{sign}(\boldsymbol{y})) = 2\int_0^1 \int_0^1 f(z_x, z_y|\rho)dz_x dz_y, \tag{8}$$

*where $f(z_x, z_y|\rho)$ is the density function of RFF given by Lemma 1.*

*Proof.* By Definition 3 and the monotonicity of $k(\rho)$, it suffices to show that the collision probability, $P_{sign}(\rho)$, is increasing in $\rho$. This immediately follows from Lemma 1 that

$$\mathbb{E}[sign(F(\boldsymbol{x}))sign(F(\boldsymbol{y}))] = P_{sign}(\rho) - (1 - P_{sign}(\rho)) = 2P_{sign}(\rho) - 1$$

is an increasing function in $\rho$. $\qquad\square$

Compared with SQ-RFF, the proposed SignRFF exhibits less "variation" brought by the stochastic sampling procedure. In fact, in the kernel approximation problem, it has been shown that stochastic rounding has higher variance due to the noise $\xi$ which hurts the kernel estimation accuracy [43]. While such comparison is not immediately obvious for the task of nearest neighbor search, next we will show that dropping the extra noise in the binary codes indeed leads to improved search performance, measured by a new metric specifically for the binary hash codes from LSH.

# 4 Ranking Efficiency (RE): A New Measure for Search Performance

In this section, we systematically compare the above LSH methods (SignRP, SQ-RFF and SignRFF) from a analytical point of view. Before we start, we first provide some analysis of KLSH introduced in Section 2 which will also be included in our comparison.

**Collision probability of KLSH.** In our theoretical analysis, we assume for KLSH [27] that the Gaussian projection in the kernel induced feature space $\mathcal{F}$ is truly random (recall in practice we use data dependent approximations). That is, KLSH "ideally" performs SignRP in the kernel space: $h_{KLSH}(\boldsymbol{x}) = sign(\boldsymbol{w}^T \Psi(\boldsymbol{x}))$ where $\Psi$ is the feature map to $\mathcal{F}$ and $\boldsymbol{w}$ is a random Gaussian vector with proper dimensionality. Since $\Psi(\boldsymbol{x})^T \Psi(\boldsymbol{y}) = k(\boldsymbol{x}, \boldsymbol{y})$, applying (2) in $\mathcal{F}$ we obtain the collision probability as

$$P_{KLSH}(\rho) := P(h_{KLSH}(\boldsymbol{x}) = h_{KLSH}(\boldsymbol{y})) = 1 - \frac{\cos^{-1}(e^{-\gamma^2(1-\rho)})}{\pi}. \tag{9}$$

**Remark 1.** *Again, we emphasize that due to the dependence among KLSH hash codes resulting from its implementation, in practice the collision probability of KLSH would be different from (9).*

**Proposition 3.** *KLSH is $(\tilde{k}, c\tilde{k}, 1 - \frac{\cos^{-1}(\tilde{k})}{\pi}, 1 - \frac{\cos^{-1}(c\tilde{k})}{\pi})$-locality sensitive w.r.t. $k(\cdot)$.*

In Figure 1, we plot the theoretical hash collision probabilities of the four hashing algorithms as in (2), (7), (8) and (9). We see that SQ-RFF has highest collision probability. Recall Definition 3 of the $(\tilde{k}, c\tilde{k}, p_1, p_2)$-LSH. It is known [21] that one can construct an LSH data structure with the worst case query time $\mathcal{O}(n^R)$, where $R := \frac{\log p_1}{\log p_2}$ is called the *LSH efficiency*, which has been used in literature, e.g., to compare SimHash vs. MinHash [59]. However, since the LSH efficiency only considers the first moment and is based on the a worst case analytical bound, it may not well explain/predict the practical search performance. See Appendix B for related discussion.

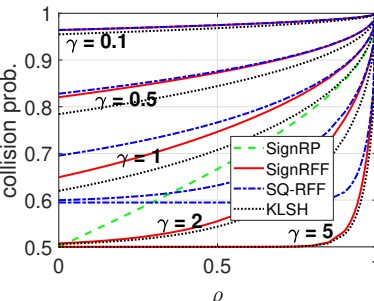

Figure 1: Hash collision probabilities.

## 4.1 Ranking Efficiency

We now introduce the concept of *ranking efficiency*, which allows us to better compare different LSH methods, theoretically, in terms of search performance. As we will see, the new metric shows the superiority of SignRFF over SQ-RFF, and provides insight on comparing (linear) SignRP with non-linear LSH methods (e.g., when is SignRFF advantageous?). The definition is motivated by the fact that the effectiveness of LSH binary codes in nearest neighbor retrieval is essentially determined by how well it can preserve the ranking of the true similarities in the Hamming space. Suppose $\boldsymbol{x}$ and $\boldsymbol{y}$ are two data points in the database, and $\boldsymbol{q}$ is a query with $\rho_x = \cos(\boldsymbol{q}, \boldsymbol{x}), \rho_y = \cos(\boldsymbol{q}, \boldsymbol{y})$. Assume $\boldsymbol{x}$ is closer than $\boldsymbol{y}$ to the query $\boldsymbol{q}$, i.e. $\rho_x > \rho_y$. By the property of LSH (Definition 3), we know that the hash collision probability $p_x > p_y$. For an LSH hash function $h$, define the corresponding collision probability estimators as

$$\hat{p}_x = \frac{1}{b} \sum_{i=1}^{b} \mathbb{1}\{h_i(\boldsymbol{x}) = h_i(\boldsymbol{q})\}, \quad \hat{p}_y = \frac{1}{b} \sum_{i=1}^{b} \mathbb{1}\{h_i(\boldsymbol{y}) = h_i(\boldsymbol{q})\}.$$

Now, the problem becomes comparing $\hat{p}_x$ and $\hat{p}_y$ to estimate the true ranking of $p_x$ and $p_y$. We consider the event of obtaining a wrong similarity comparison from our estimation, i.e. $\hat{p}_x \leq \hat{p}_y$. Obviously, a higher probability implies worse search accuracy, as we are more likely to retrieve the wrong neighbor $\boldsymbol{y}$. Denote $E_x = \mathbb{E}[\mathbb{1}\{h(\boldsymbol{x}) = h(\boldsymbol{q})\}], E_y = \mathbb{E}[\mathbb{1}\{h(\boldsymbol{y}) = h(\boldsymbol{q})\}]$, and $Cov(\mathbb{1}\{h(\boldsymbol{x}) = h(\boldsymbol{q})\}, \mathbb{1}\{h(\boldsymbol{y}) = h(\boldsymbol{q})\}) = \Sigma = \begin{pmatrix} V_x & V_{xy} \\ V_{xy} & V_y \end{pmatrix}$. By the Central Limit Theorem, as $b \to \infty$ asymptotically, we have that $\begin{pmatrix} \hat{p}_x \\ \hat{p}_y \end{pmatrix} \sim N(\begin{pmatrix} E_x \\ E_y \end{pmatrix}, \Sigma/b)$. This approximation would be good with a sufficiently large $b$, e.g. $b \geq 30$. In this regime, we have

$$P(\hat{p}_x \leq \hat{p}_y) = P(\hat{p}_x - \hat{p}_y \leq 0) = 1 - \Phi\left(\frac{\sqrt{b}(E_x - E_y)}{\sqrt{V_x + V_y - 2V_{xy}}}\right), \tag{10}$$

where $\Phi(\cdot)$ is the c.d.f. of standard normal distribution. Based on this characterization, we formally define the *ranking efficiency* as follows.

**Definition 4** (($\rho, c$)-Ranking Efficiency (RE)). *For a LSH method, let the hash collision probability at cosine $\rho$ and $c\rho$ be $E$ and $E_c$, respectively, with $0 \leq c < 1$. Let $V = E(1 - E)$, $V_c = E_c(1 - E_c)$. The $(\rho, c)$-ranking efficiency (RE) is defined as*

$$\textit{Ranking Efficiency:} \quad RE = \frac{E - E_c}{\sqrt{V + V_c}}. \tag{11}$$

**Remark 2.** *In most cases, we may care more about large $c$ values (e.g., $c = 0.95$) which corresponds to similar data points with higher chance of reversed ranking.*

**Remark 3.** *The covariance $V_{xy}$ in (10) is in general intractable which depends on the specific data $\boldsymbol{x}$ and $\boldsymbol{y}$. We assume that it has same relative impact on the RE values and drop it from the definition for simplicity. The RE, when multiplied by $\sqrt{b}$, is equivalent to the z-score in a two-sample z-test.*

Higher $RE$ implies smaller probability of (10), which is more favorable. For all the LSH methods, the $E_x$ and $E_y$ are concretely the collision probabilities given in (2), (9), (7) and (8), and $V_x = E_x(1 - E_x)$, $V_y = E_y(1 - E_y)$ from the binomial distribution, respectively.

## 4.2 Analytical Comparison of LSH Methods Through Ranking Efficiency

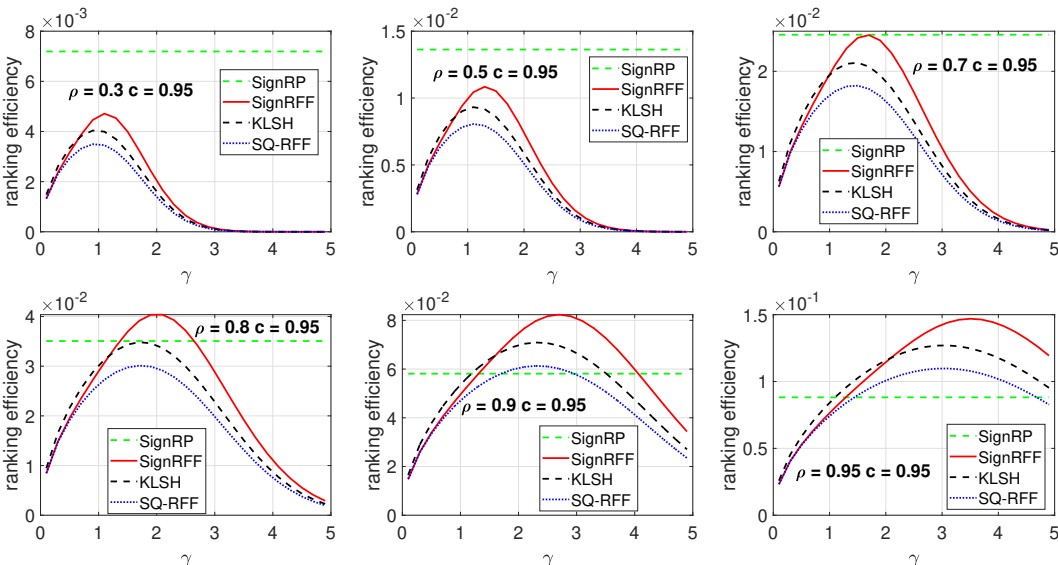

Figure 2: Ranking efficiency (Definition 4) of different LSH methods with various $\rho$, at $c = 0.95$.

Next, we leverage Definition 4 to theoretically compare different hashing methods. In Figure 2, we provide the RE of (linear) SignRP, KLSH, SQ-RFF and SignRFF at different $\rho$ level, which covers the cases in our empirical study (Section 5). We present the plots for $c = 0.95$, and the conclusions are the same for other $c$ values. We observe that:

- **Comparing Non-linear LSHs.** Firstly, compared with the previous RFF-based approach SQ-RFF, the proposed SignRFF is uniformly more efficient at all $\rho$, verifying its superiority. Compared with KLSH, SignRFF is more efficient when $\gamma$ is large (e.g., $\gamma > 2$) in all plots, while KLSH tends to be more efficient only with small $\gamma$.

- **When should we prefer non-linear LSH?** We see that when $\rho$ is high (e.g., $\rho > 0.8$), with proper tuning, kernel methods (e.g., SignRFF) can be more efficient than SignRP. However, if the target $\rho$ is small (e.g., $\rho < 0.7$), SignRP becomes more favorable, even for best tuned non-linear LSH. In other words, SignRFF is better than SignRP on datasets where the near neighbors are close to each other, with high similarity/short distance.

The ranking efficiency measures the search accuracy for a given $\rho$ level. In practice, a dataset contains many sample pairs with different $\rho$. Our experiments in the next section show that the performance

is largely consistent with the prediction at the "average $\rho$" level. That said, ranking efficiency may provide a convenient way to choose the proper LSH method based on the data of interest.

## 5   Experiments

We conduct experiments to demonstrate the effectiveness of our approach and justify that ranking efficiency indeed provides reliable prediction of the empirical search accuracy.

**Datasets.** We use three popular benchmark datasets for image retrieval. The SIFT dataset [23] contains 1M 128-dimensional SIFT image features, and 1000 query samples. The MNIST dataset [29] contains 60000 hand-written digits. The CIFAR dataset [25] contains 50000 natural images and we use the gray-scale images in our experiments. For these two datasets, we randomly choose 1000 samples from the test set as queries. In addition, when comparing with VGG-16 [60] based deep methods, following prior literature (e.g.,[69, 52]), we use the 4096-d features from the last fully connected layer of the pre-trained VGG-16 network as the input data for shallow methods for fairness. This dataset is called "CIFAR-VGG". On all datasets, the data points are normalized to have unit norm. In Figure 3, we report the average cosine between queries to their $N$-th nearest neighbor, which can be approximately regarded as the "target $\rho$ level" when we compare the ranking efficiency (e.g., 0.95 for CIFAR).

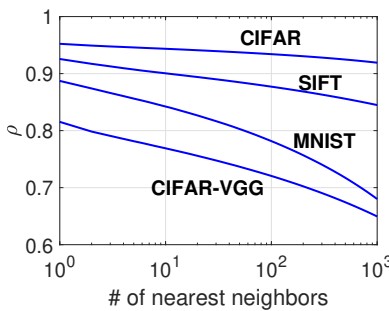

Figure 3: Average $\rho$ to the $N$-th neighbor on four datasets.

**Methods and Evaluation.** We compare the following unsupervised hashing methods: 1) **SignRP** [5], defined by (1) with random Gaussian projections; 2) **Iterative Quantization (ITQ)** [17], which finds rotations to minimize the quantization error of mapping the randomly projected data to binary; 3) **Spectral Hashing (SpecH)** [65], which is based on quantizing the values of analytical eigenfunctions computed along principal component directions of the data; 4) **Binary Reconstruction Embedding (BRE)** [26], which explicitly minimizes the reconstruction error between the original distances and the Hamming distances. We use 1000 random samples for model training as suggested by [26]; 5) **KLSH** [27] as in (6), with $m = 500$ random samples for formulating the kernel matrix and $t = 50$ samples for the CLT Gaussian approximation, more accurate than $(300, 30)$ recommended in [27]; 6) **SQ-RFF** [53] given in (5), binary codes from stochastically quantized RFF; 7) Our proposed **SignRFF** method (4) with deterministic quantization from RFF. For methods (5) - (7) involving Gaussian kernel, we tune $\gamma$ on a fine grid over $0.1 \sim 5$ and report the best result.

For each tested method, we generate binary codes and find neighbors based on Hamming distances. After running each algorithm, the Hamming ranking returns $R$ estimated nearest neighbors to each query. Define $N$ as the number of ground truth neighbors. For each query point, the ground truth nearest neighbors are set by ranking the top $N = 100$ smallest Euclidean distance (top-100 largest cosine similarity). We report the average recall or precision over 1000 queries. The search recall and precision (the higher the better) are defined as recall@R $= \frac{\text{\# true neighbors in } R \text{ retrieved points}}{N}$ and precision@R $= \frac{\text{\# true neighbors in } R \text{ retrieved points}}{R}$. Note that recall@N is equivalent to precision@N.

### 5.1   Results

In Figure 4, we report the recall@100 (precision@100) against the number of binary codes $b$:

- On SIFT and MNIST, data-dependent methods (ITQ, SpecH, BRE) perform well with low bits, but the recall does not improve much after $b > 100 \sim 200$. Yet, their recall level is too low (e.g. $< 0.3$ on SIFT and CIFAR) for real-world tasks, which is a known limitation of these methods. When $b \geq 256$, LSH-type methods start to dominate. On CIFAR, SignRFF and KLSH outperform all the data-dependent methods even with low bits.

- On all three datasets, SignRFF is substantially better than SQ-RFF with all $b$. Due to dependence, KLSH has higher recall than SignRFF when $b$ is small (e.g., $\leq 256$), but is beaten by SignRFF with more bits. The gap is significant and consistent when $b$ is as large as 512, where SignRFF achieves the highest recall on all three datasets.

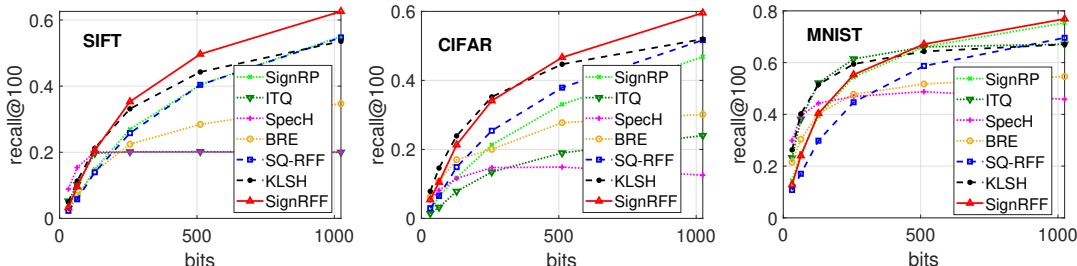

Figure 4: Recall@100 vs. $b$. Note that in our case, recall@100 is equivalent to precision@100.

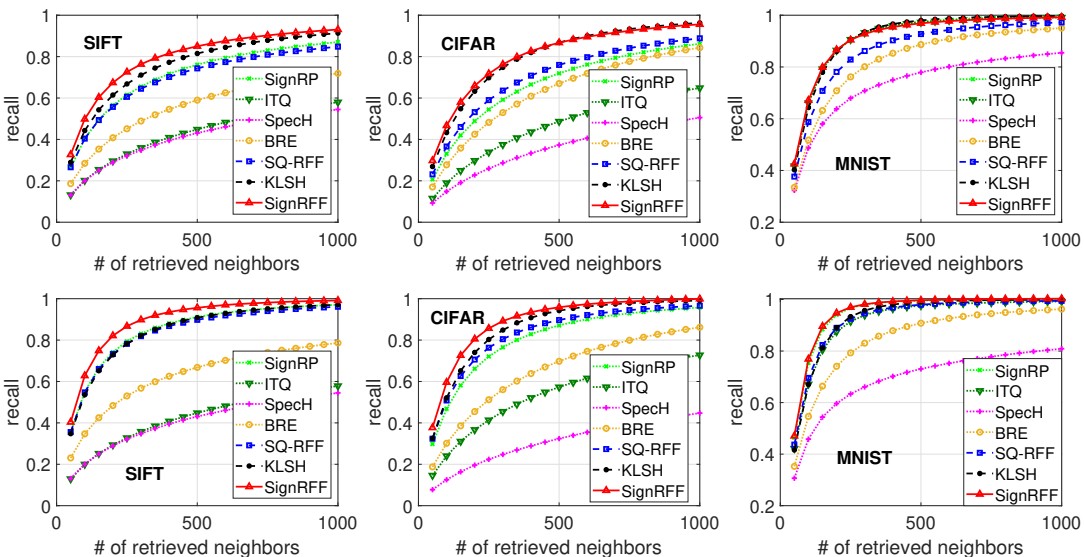

Figure 5: Recall@R vs. # of retrieved neighbors $R$. **1st row:** $b = 512$. **2nd row:** $b = 1024$.

In Figure 5, we present the recall versus the number of retrieved neighbors, with $b = 512$ and $b = 1024$. In general, SignRFF performs the best on all three datasets with a sufficient number of bits. Due to space limitation, we report search precision results with consistent conclusions in Appendix A, which also includes discussion on the practical implementation and efficiency.

**Comparison with Deep Hashing.** We provide experiments on the CIFAR-VGG dataset using a recent unsupervised deep hashing method, the Contrastive Information Bottleneck (CIB) [52], which uses VGG-16 pre-trained model as the backbone. We apply the same training setting as in [52]. Note that, CIB is actually not designed to find the most similar data points. Instead, by using techniques like cropping and rotation in CNN, CIB aims at finding data points with the same label as the query, which does not necessarily imply high similarity. Hence, to favor CIB, in this experiment we expand the range of true neighbors of a query to the top-1000 most similar points in the database as in [52] (see Appendix C for detailed discussion). From the results in Figure 6, we observe the following:

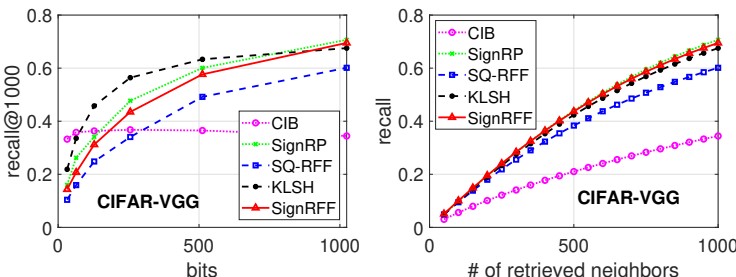

Figure 6: **Left:** Recall@1000 (precision@1000) vs. $b$. **Right:** Recall vs. # retrieved points, $b = 1024$.

- From Figure 6, CIB performs the best when $b \leq 64$, illustrating the benefit of deep hashing with short codes. Yet, the recall is only 0.3∼0.4, and does not improve with more bits. When $b \geq 256$, SignRP, SignRFF and KLSH provide much higher recall than CIB.
- Due to the dependence among codes, KLSH performs the best on this dataset with short codes. SignRFF again consistently improves SQ-RFF, and surpasses KLSH with $b = 1024$.

## 5.2 Ranking Efficiency is a Predictive Measure of Search Accuracy

Firstly, as shown in Figure 3, the average $\rho$ between each query and its near neighbors is around 0.7, 0.8, 0.9 and 0.95 for `CIFAR-VGG`, `MNIST`, `SIFT` and `CIFAR`, respectively. The theoretical RE in Figure 2 suggests that compared with SQ-RFF (blue), SignRP (light green) would perform better on `MNIST` and `CIFAR-VGG`, similarly on `SIFT`, and worse on `CIFAR`. This aligns very well with the recall curves in Figure 4 and Figure 5. Here we use SQ-RFF vs. SignRP as an example; similar alignment holds for comparing other methods. To make more detailed justification, in Figure 7, we additionally provide the recall against the Gaussian parameter $\gamma$. In Figure 2, the RE curves predict that KLSH should perform better with small $\gamma$, while SignRFF is more powerful with larger $\gamma$. This matches the empirical evidence, e.g., 1 for KLSH vs. 2.5 for SignRFF on `SIFT` to reach highest recall.

Our results verify that: 1) the ranking efficiency can be used as an informative and effective measure to compare different LSH methods in practice; 2) kernel based LSH (e.g., the proposed SignRFF) is more favorable than the linear SignRP method in high similarity region.

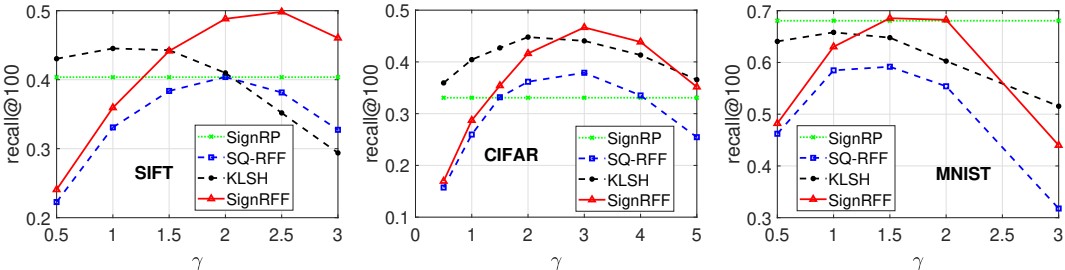

Figure 7: Recall@100 (precision@100) vs. $\gamma$, with $b = 512$.

## 6 Conclusion

In this paper, we develop **SignRFF** (sign random Fourier features) as an alternative to the popular method of SignRP (sign random projections). SignRFF and SignRP fit in the pipeline of embedding based retrieval (EBR) quite smoothly. EBR has become the standard industry practice. SignRFF can be applied after the embedding vectors have been trained to serve multiple purposes. It can be used as data reduction tool for efficiently storing embeddings and computing similarities. It can also be used an indexing scheme for approximate near neighbor (ANN) search, which is a crucial component in EBR. To assess the quality of SignRFF in the context of similarity rankings, we design a new unified theoretical framework to compare different hashing methods, based on a measure called **Ranking Efficiency (RE)**. In terms of RE, we show that SignRFF outperforms SQ-RFF and KLSH which are existing methods related to RFF. In the relatively high similarity region, SignRFF also outperforms SignRP in terms of the RE measure. The theoretical findings are supported by experiments on datasets commonly used in the literature. Comparisons with deep learning as well as data-dependent hashing methods are also provided, to further confirm the effectiveness of SignRFF.

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
