# A More Experiments

## A.1 Precision

In Figure 8, we report the Precision@10 against the number of bits (1st row) and precision@R against the number of retrieved points $R$ (2nd row). Basically, we get the same conclusions as those from the recall curves. SignRFF performs the best on all datasets after $b \geq 256$.

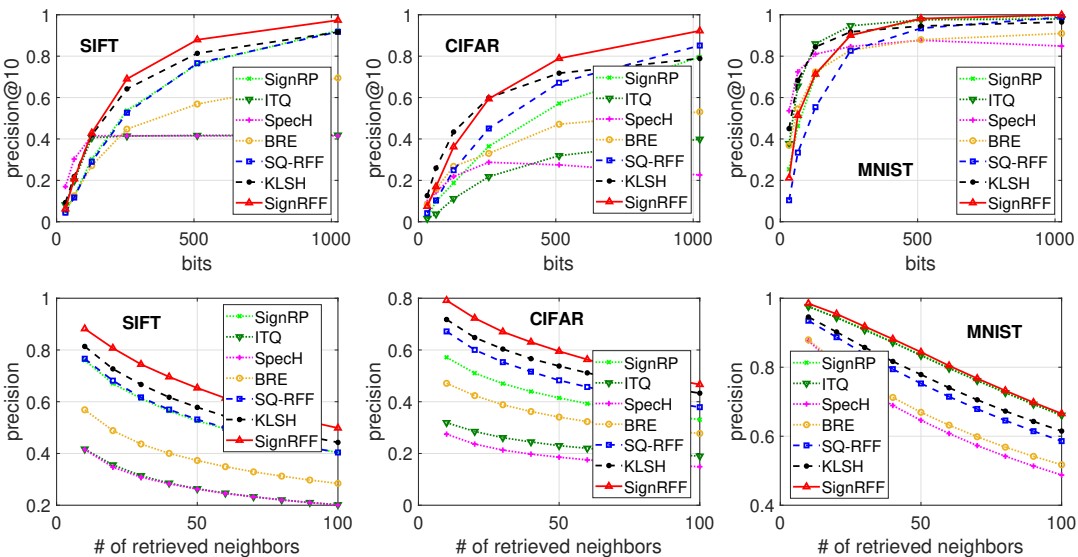

Figure 8: **1st row:** Precision@10 against $b$. **2nd row:** Precision vs. # of retrieved points, $b = 512$.

## A.2 Practical Considerations and Efficiency

In practical retrieval systems, the query time is an important consideration which consists of the processing time (to generate query binary codes), and the search time (to compare the query codes to the database). We briefly discuss the two aspects below.

**Processing time.** Noticeably, another benefit of SignRFF is its simplicity in implementation. This can be reflected in the data processing time. In Figure 9, we plot the comparison of processing time (for 1000 queries). We observe that SignRFF and LSH are the most efficient methods (mostly only a random projection). KLSH is notably slower than SignRFF. The two data-dependent methods, SpecH and BRE, are significantly slower than SignRFF. This reveals a potential advantage of the simplicity of SignRFF in practical retrieval systems.

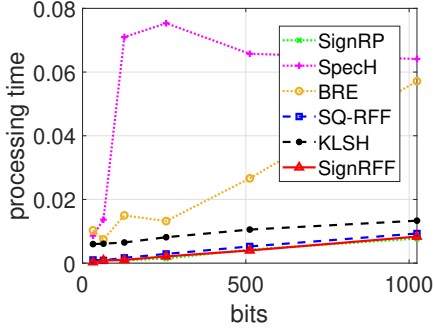

Figure 9: Data processing time (1000 queries).

**Search time.** In our experiments, we adopt the standard exact Hamming search by linear scan. On a single core 2.0GHz CPU compiled with C++, searching over 1M samples on SIFT takes

approximately 0.15s per query with $b = 512$. Note that linear scan is a naive strategy. While the efficiency of Hamming search algorithms is not the focus of this work, we point out that in practice, we can use multi-index hashing [50, 51] to perform exact Hamming search with a substantial $10^2 \sim 10^3$ times acceleration. Additionally, the method in [51] is particularly effective for long codes, the regime where SignRFF possesses its best advantage. As such, searching with SignRFF can also be very efficient in practice.

# B More Analytical Figures

## B.1 LSH Efficiency is Not Enough to Predict Search Accuracy

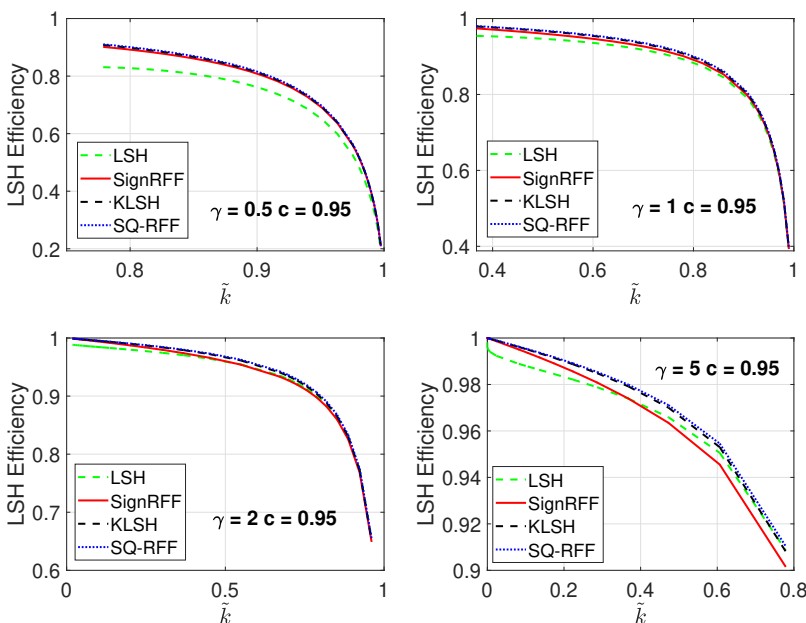

Figure 10: LSH efficiency of different LSH methods with various $\gamma$, $c = 0.95$. The $x$-axis $\tilde{k}$ is the Gaussian kernel value in alignment with Definition 3, Proposition 1, 1 and 3.

Recall Definition 3 of the $(\tilde{k}, c\tilde{k}, p_1, p_2)$-LSH. It is known [20] that one can construct an LSH data structure with the worst case query time $\mathcal{O}(n^R)$, where $R := \log p_1 / \log p_2$ is called the *LSH efficiency*, which has been used in literature to theoretically compare different LSH methods, e.g. SimHash vs. MinHash [59]. However, we found that in our case, the LSH efficiency does not provide much informative comparison of different LSH methods of interest. In Figure 10, we provide the LSH efficiency at multiple $\gamma$. Firstly, we see that the differences among the curves are very small. Basically, the figures tells us that SignRFF is always better than KLSH and SQ-RFF, but do not provide the comparison of SignRFF and KLSH regarding $\gamma$, as the ranking efficiency does. Also, the figures seem to suggest that SignRFF could (roughly) be better than LSH with large $\gamma$ and large $\rho$, but it does not give a good threshold at around $\rho = 0.7$ (validated by the experiments) as suggested by the ranking efficiency. Thus, LSH efficiency is insufficient to well predict the practical Hamming search performance.

## B.2 Ranking Efficiency: More $c$ and $\rho$ Values

we provide more theoretical comparisons on the ranking efficiency at more $\rho$ and $c$ values. The observation (relative comparison) is similar to the results presented in the main paper. For convenience, we plot the ratio of ranking efficiency of KLSH and SQ-RFF over SignRFF. In Figure 11, Figure 12 and Figure 13, we plot the ratio at a wide range of $\rho = 0.01 \sim 0.95$ and $c = 0.95, 0.7, 0.5$. We see that the ratio SignRFF/SQ-RFF is always larger than 1, i.e., SignRFF is always more efficient than SQ-RFF. At small $\gamma$, KLSH is more efficient (ratio SignRFF/KLSH less than 1), while for larger $\gamma$, SignRFF is better. This is consistent with the result presented in the main paper.

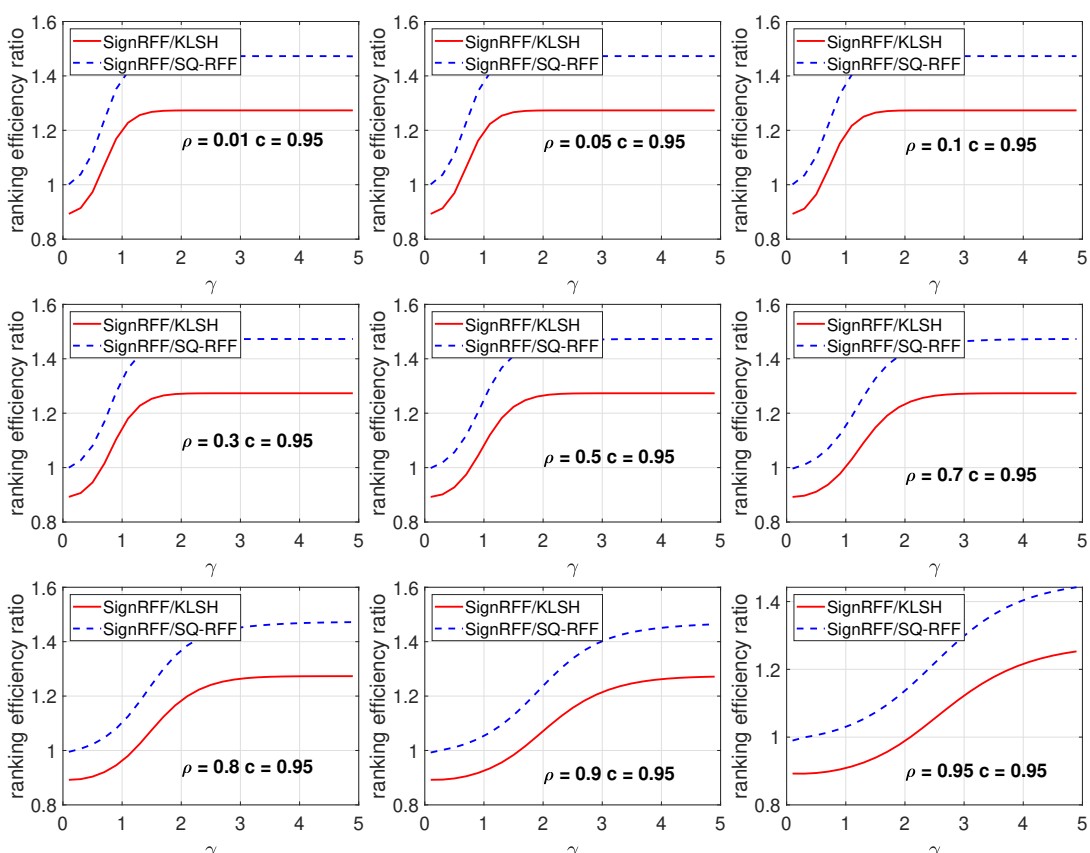

Figure 11: Ranking efficiency ratio against Gaussian kernel parameter $\gamma$ at various $\rho$. $c = 0.95$.

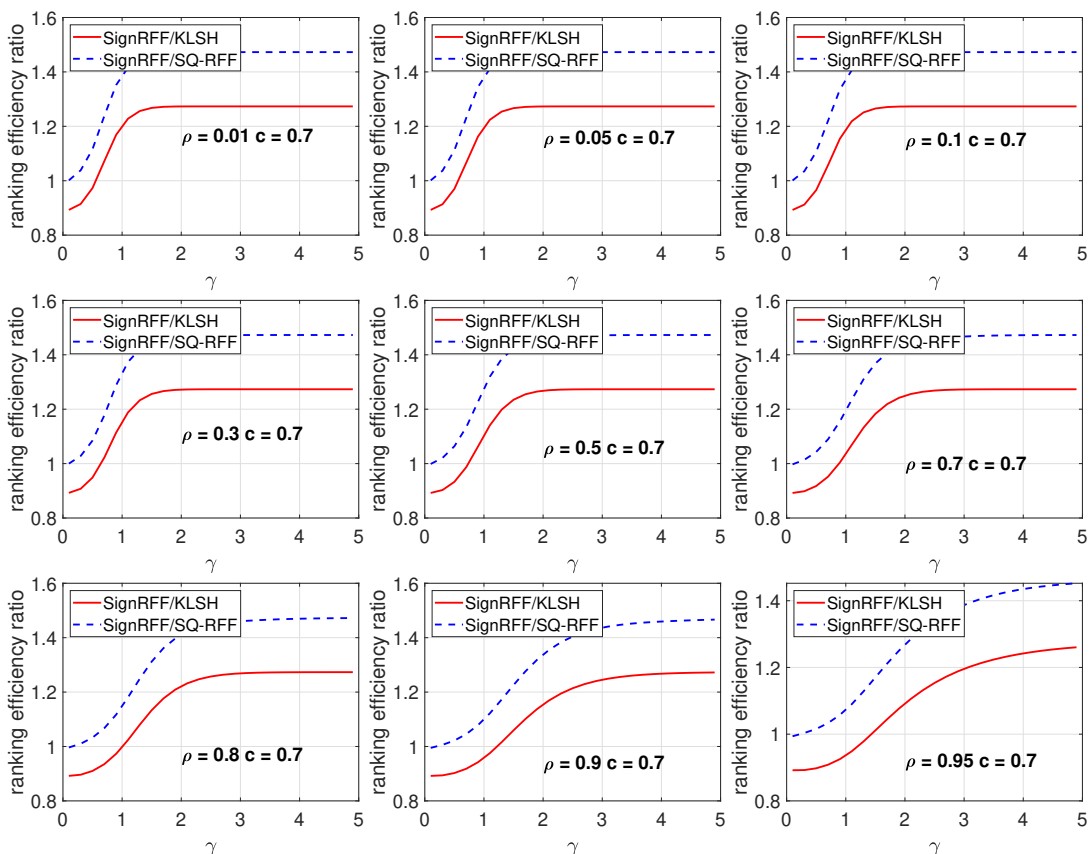

Figure 12: Ranking efficiency ratio against Gaussian kernel parameter $\gamma$ at various $\rho$. $c = 0.7$.

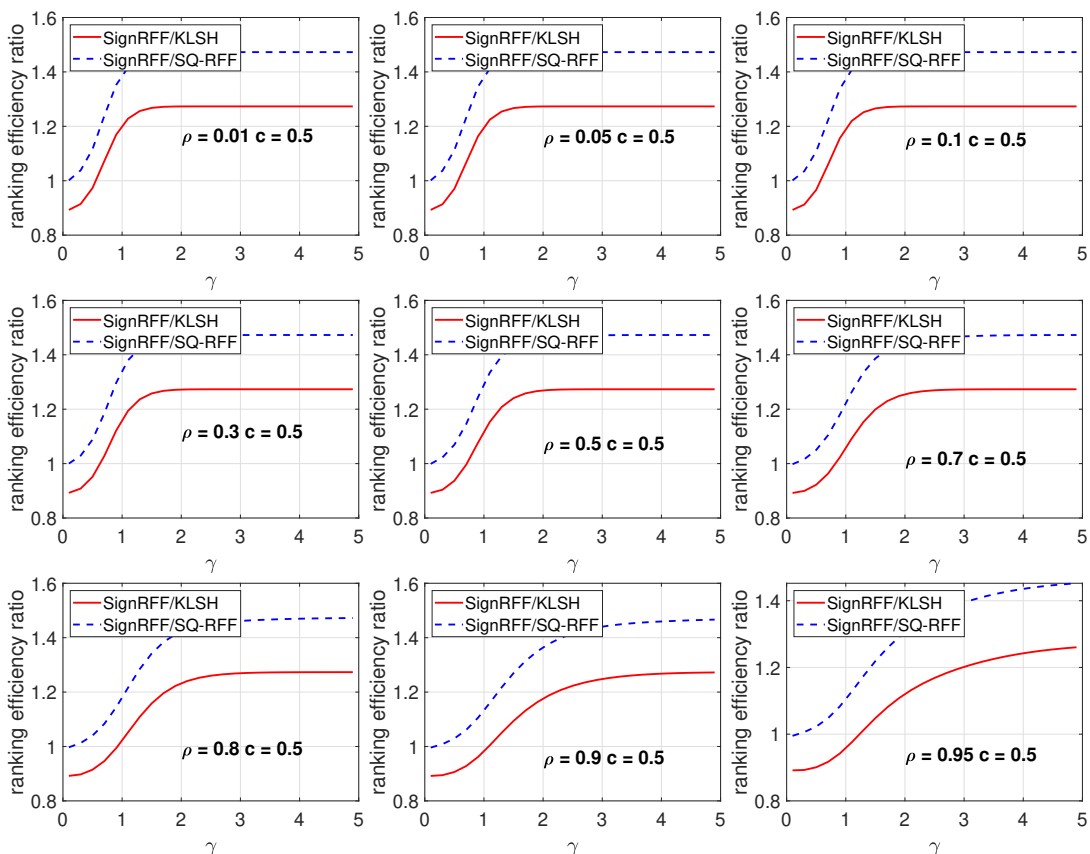

Figure 13: Ranking efficiency ratio against Gaussian kernel parameter $\gamma$ at various $\rho$. $c = 0.5$.

## C    More Image Retrieval Results on `CIFAR-VGG`

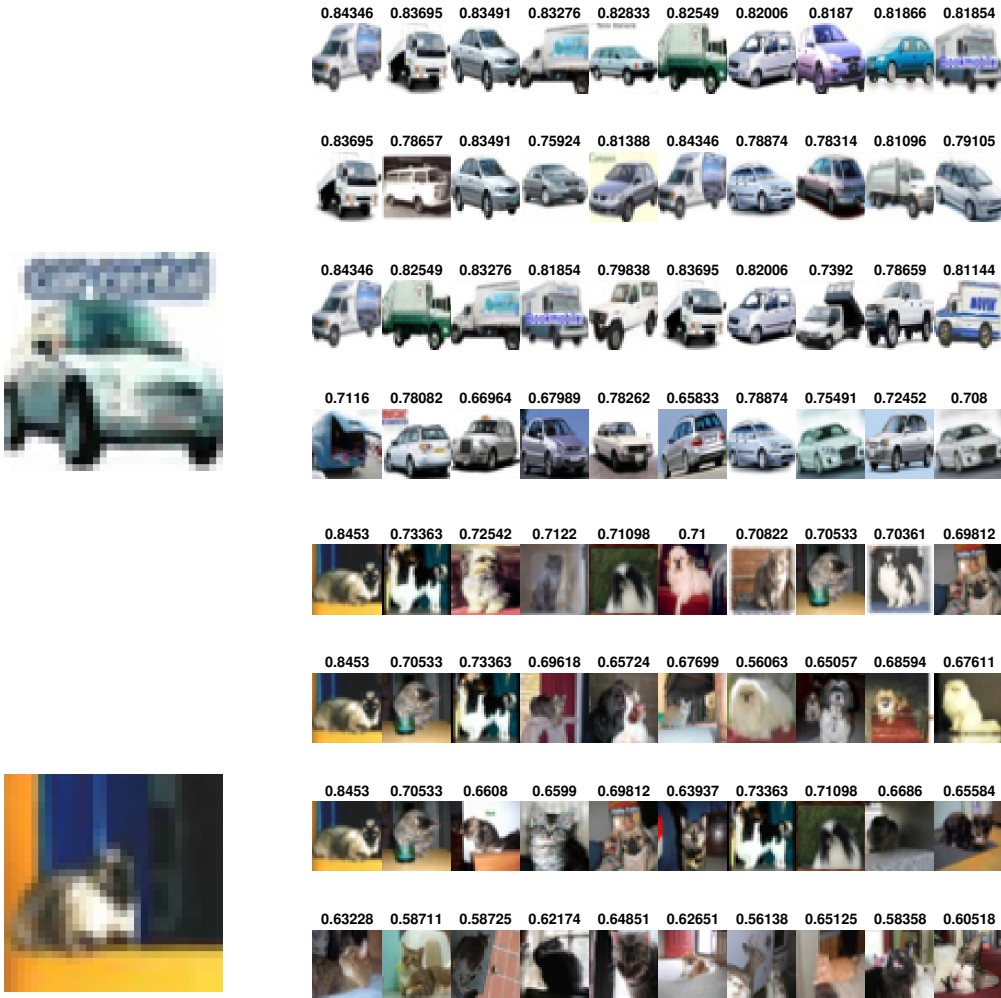

Figure 14: `CIFAR-VGG` Top-10 retrieved images (right) for two example query images (left, automobile and cat) with $b = 512$. **1st row:** true nearest neighbors in terms of cosine similarity of the features extract from the last fc layer of VGG-16. **2nd row:** SignRFF. **3rd row:** KLSH. **4th row:** CIB. The number on each retrieved image is the cosine similarity of its VGG-feature to the VGG-feature of the query image.

In Section 5.1, we compare the locality-sensitive hashing methods with a deep learning based method, the Contrastive Information Bottleneck (CIB) [53]. CIB is built upon the pre-trained VGG-16 CNN model, whose objective is to generate binary representations such that images from the same class are close. Thus, same as many other deep hashing methods, the objective of CIB is slightly different from our setting (we find the most similar data points, without any label information), as being in the same class does not necessarily implies high similarity. In fact, in the empirical evaluation of many of these papers (e.g., [69, 53]), the true neighbors are simply set as those data points with same label as the query. In our setting, we strictly follow the ranking to find the true neighbors with highest similarities. To better illustrate the difference, we present top-10 retrieval result of two queries (automobile and cat) in Figure 14. For SignRFF and KLSH, the retrieved images mostly have high similarity with the query, but may include some data points from other classes (e.g., truck vs. automobile, dog vs. cat). On the contrary, the retrieved images of CIB clearly have lower similarity with the query, but contain fewer other classes. This is largely because the VGG-16 is pre-trained by classification. Hence, in

this experiment we set the true neighbors as the top-1000 similar data points, following the original paper [53], where CIB could be evaluated properly since most images from the same class would, at least, have not too low similarity.

# D  Impact of the Dependence in KLSH

In practical implementation, the hash codes of KLSH are dependent. We give an intuitive explanation on how this dependence affect the search performance. Firstly, let us review the reason why data-independent methods with independent codes (LSH, SQ-RFF, SignRFF) can boost search accuracy with increasing $b$. Essentially, same as the intuition of the rank efficiency, comparing Hamming distance is equivalent to searching for the data points with highest estimated hash collision probability using $b$ codes. Let $\boldsymbol{x}, \boldsymbol{y}$ be two database points and $\boldsymbol{q}$ be the query, with $\rho_x > \rho_y$. This means the hash collision probability $p_x > p_y$. We estimate the probability by averaging the collision indicators:

$$\hat{p}_x = \frac{1}{b} \sum_{i=1}^{b} \mathbb{1}\{h_i(\boldsymbol{x}) = h_i(\boldsymbol{q})\}, \quad \hat{p}_y = \frac{1}{b} \sum_{i=1}^{b} \mathbb{1}\{h_i(\boldsymbol{y}) = h_i(\boldsymbol{q})\}. \tag{12}$$

For data-dependent methods, with sufficient $b$, (10) gives the probability of wrong ranking (i.e., $\hat{p}_x < \hat{p}_y$), strictly decreasing with $b$.

Recall the steps of KLSH. We first sample $m$ data points from database $\boldsymbol{X}$, denoted as $\tilde{\boldsymbol{X}}$, to form a kernel matrix $\boldsymbol{K}$. Then we uniformly pick $t$ points, denoted as $\tilde{\boldsymbol{X}}'$, from $[1, ..., m]$ at random to approximate the Gaussian distribution. After some algebra, the hash code has the form

$$\textbf{KLSH:} \quad h(\boldsymbol{x}) = sign(\sum_{i=1}^{m} \boldsymbol{w}(i) k(\boldsymbol{x}, \boldsymbol{x}_i)), \tag{13}$$

where $\boldsymbol{w} = \boldsymbol{K}^{-1/2} \boldsymbol{e}_t$, and $\boldsymbol{e}_t \in \{0, 1\}^m$ has ones in the entries with indices of the $t$ selected points. To generate multiple codes, we use the same pool of points $\tilde{\boldsymbol{X}}$, but with different choice of $e_t$, i.e., $\tilde{\boldsymbol{X}}'$ to approximate the Gaussian distribution.

**Boosted performance with small $b$.** For two hash codes $h_k$ and $h_{k-1}$ of $\boldsymbol{x}$, the $k(\boldsymbol{x}, \boldsymbol{x}_i)$ terms in (13) are the same. The $\boldsymbol{w}$'s are dependent since the points used for Gaussian approximation (i.e. $\tilde{\boldsymbol{X}}'$) may overlap. Thus, the conditional hash collision probability is usually larger than the unconditional one,

$$P(h_k(\boldsymbol{x}) = h_k(\boldsymbol{y}) | h_{k-1}(\boldsymbol{x}) = h_{k-1}(\boldsymbol{y})) > P(h_k(\boldsymbol{x}) = h_k(\boldsymbol{y})),$$

and it may increase as $k$ get larger (intuitively, by dependence, more previous collisions implies higher chance of collision later on). Similarly,

$$P(h_k(\boldsymbol{x}) \neq h_k(\boldsymbol{y}) | h_{k-1}(\boldsymbol{x}) \neq h_{k-1}(\boldsymbol{y})) > P(h_k(\boldsymbol{x}) \neq h_k(\boldsymbol{y})).$$

Thus, at the beginning, the more similar (dissimilar) the data pair is, the more the estimator $\hat{p}$ is upward (downward) biased. In other words, similar points would have further increased hash collision probability, while dissimilar points would have even lower chance of collision. This is the main reason that the empirical performance of KLSH is higher than theoretical prediction (where we assume independence) with short binary codes.

**Slow improvement with large $b$.** However, this is not the whole story. As more bits are used, there is less and less marginal information left. That is, the later generated codes would be more and more dependent on the previous ones. In an extreme case, at the point when all $\binom{m}{t}$ combinations of the $t$ points in $\tilde{\boldsymbol{X}}'$ have been used, the hash codes produced afterwards would all be the same as some previously generated ones, which would hardly improve the distance estimation anymore. This is why for KLSH, the recall curve becomes flat as $b$ increases.