# OpenReview forum: "SignRFF: Sign Random Fourier Features"
_NeurIPS.cc/2022/Conference — NeurIPS 2022 Accept_

### Official Review · Reviewer_bnZk · 2022-07-10

**Rating:** 7
**Confidence:** 3
**Soundness:** 3 good
**Presentation:** 3 good
**Contribution:** 3 good

**Summary:**

This paper studies the problem of binary code constructions, which can be used to approximate the original high-dimensional vectors of real numbers. The approximation can save storage space and accelerate similarity search runtime. The paper introduces a new method, called SignRFF and claims the following contributions: (1) better performance than extensive baselines (i.e. LSH, ITQ, SpecH, BRE, SQ-RFF, and KLSH) (2) introduces a new metric called ranking efficiency to evaluate the quality of binary codes.

**Questions:**

Address the weaknesses

**Limitations:**

N.A.

**Strengths And Weaknesses:**

Strengths:

S1: Novel technical contribution. The paper targets at SQ-RFF algorithm, which is becoming a standard for binary code construction and has over 750 citations since 2010. The paper reveals that the random noise component of the SQ-RFF algorithm is unnecessary, and it proposes a new algorithm SignRFF to remove the component. Experiments verify that SignRFF is more effective than SQ-RFF.

S2: Solid experiments. The paper compares SignRFF with not only SQ-RFF but also a set of other strong baselines, including LSH, ITQ, SpecH, BRE, SQ-RFF. These baselines have broad impact in the field. For example, both LSH and ITQ have gained more than 1000 citations since the works were published.

Weaknesses:

W1: Unclear motivation about new metric. The paper introduces a new metric called ranking efficiency to evaluate the quality of binary codes. The new metric is a key contribution claimed by the authors. It is unclear why we need a new metric. There are already a set of metrics commonly used in the field, including recall@#bits or recall@topk. It will be convincing if the paper explains the motivation and how the new metric differs from common ones.

W2: no deep hashing methods in key experiments. Figures 3 and 4 show that the proposed method outperforms a number of baselines. But none of the baselines is deep learning based. The paper moves comparison with deep hashing (i.e. CIB) to Figure 5, where the metric differs from Figures 3 and 4. It will be more convincing if the papers can add the performance numbers of CIB to Figures 3 and 4.

---

> ### Author Response · Authors · 2022-08-02
> **Response to Reviewer bnZk**
>
> We sincerely appreciate your support and encouraging comments on our technical and empirical contributions.
>
> W1. The new metric of ranking efficiency (RE) is introduced so that we can theoretically compare different methods based on the statistical properties of the hashing algorithms. We agree that empirical measures such as precision and recall are practically effective and our experiments also adopted precision and recall for the  empirical comparisons.
>
> In this paper, we hope to provide a deeper understanding why the proposed method (SignRFF) works well, not just empirically (based on precision and recall). The primary motivation of RE, is to characterize the search performance in terms of the "reverse ranking probability", as given in Equation (10). The RE curves in Figure 1 are derived only using the formulas (2), (9), (6), (8), without being tested on any dataset. It turns out that (Section 5.2), the theoretical RE comparisons are highly consistent with the empirical search performance, which gives several main conclusions:
>
> (i) SignRFF is always better than SQ-RFF;
>
> (ii) Non-linear LSH methods (e.g., SignRFF) can be better than linear LSH in high similarity region;
>
> (iii) Theoretical RE is a predictive measure of empirical search accuracy.
>
> Note that, conclusions (i) and (ii) are novel, and not only empirical, but more importantly also supported by the theory of ranking efficiency. In practice, RE may help decide which LSH to use based on the data statistics (e.g., Figure 2), waving the need to run an extra cross-validation to choose the proper method.
>
>
> W2. Now we believe it is more clear that the primary topic of our work is to improve the SQ-RFF method and theoretically compare different LSH family algorithms (by RE), which are extensively justified in Section 5. The main purpose of including deep hashing results is to show that deep hashing can be better than classical data-dependent hashing methods, but still suffer severely with moderately long codes (e.g., when $b>64$). Therefore, we provide this additional comparison with one of the latest unsupervised deep hashing method CIB [34] (published in 2021) in the paper. In [34], the authors have shown the advantage of CIB over many more deep hashing methods. Thus, we feel that our results in Section 5.1 might be sufficient for the purpose of our work, but please let us know if there are better (and more recent) baseline than CIB [34].
>
> We explained the reason for using a different metric for CIB (recall@1000) at line 296-301 and in Appendix C. In short, this is because deep hashing usually targets at a slightly different objective, to find samples in the same class (using CNN techniques), but not the most similar ones (see Figure 13 in the appendix for example). That is why we used larger recall range for evaluation (this actually favors CIB). We use the specific recall@1000 metric following the original CIB paper [34].
>
> Again, we appreciate your support and valuable feedback.

---

### Official Review · Reviewer_HPJi · 2022-07-11

**Rating:** 4
**Confidence:** 4
**Ethics Flag:** Yes
**Soundness:** 2 fair
**Presentation:** 3 good
**Contribution:** 2 fair

**Summary:**

This paper modified the LSH method on RFF feature in [35] by removing the random noise and prove that it is still a LSH. Moreover, this paper also provides a new metric ranking efficiency to evaluate hash codes. Experiments are conducted on multiple data sets like SIFT, CIFAR, MNIST, comparing the proposed method to original one in [35] as well as other unsupervised hashing methods and one deep hashing method.

**Questions:**

--A more rigorous analysis about why $V_{xy}$ can be discarded.
--Compared to more SOTA unsupervised deep hashing methods.
--Evaluate the codes with multi hash tables besides hamming ranking.

**Limitations:**

I did not find the discussion of the technical limitation or potential negative societal impact, even though the authors claims they are discussed.

**Strengths And Weaknesses:**

Strengths:

--The theoretical part of the proposed modification looks sound.

--The experiments show the proposed modification outperforms the original one in [35].

Weaknesses:

--To prove the proposed SignRFF is LSH, the major technical difficulty (and hence contribution) is to formulate $f(z_x, z_y | \rho)$ and prove  it is increasing with $\rho$, which, however, is borrowed from [26]. So the technical contribution is not high.

--To get the ranking efficiency metric, $V_{xy}$ is discarded from equation (10). However, is it reasonable?  Is $V_{xy}$ close to 0, or very small, compared to $V_{x}$ and $V_{y}$, at least under some conditions? It is  is not rigorous to discard the term simply because it is hard to derive.

--There are many unsupervised deep hashing methods, and only one of them is compared.

--Only hamming ranking with long codes is used to evaluate the hash codes in this paper. However, the real practical way to use hash codes is actually constructing multi hash tables, each with a short code, which is how the sub-linear performance of LSH can be achieved, (and also how the complexity bound is proved).

---

> ### Author Response · Authors · 2022-08-02
> **Response to Reviewer HPJi**
>
> Thanks for your valuable comments. We have include related results and discussion regarding your questions in the revision.
>
> 1. Indeed, the covariances are not very small but in our case discarding this term did not affect the results of comparisons (between SignRFF and SQ-RFF). We discarded this term only because it could not be (easily) calculated for KLSH and we hoped to include KLSH in the comparisons. Initially, when we wrote the paper, we had figures like Figure 9, 10, and 11 in the Appendix (of the revision) to plot the "full RE"  $\frac{E-E_c}{\sqrt{V+V_c-2V_{,c}}}$. But in order to include KLSH, we decided to define $RE =   \frac{E-E_c}{\sqrt{V+V_c}}$ and presented Figure 1. We agree there is no strong reason to discard the covariance term, at least it could be placed in the Appendix.
>
> As shown in Figures 9 and 10, We justify that the shapes of the curves and the relative comparisons are essentially the same as those of the "simplified" RE. Roughly, adding the covariance term essentially up-scales the value. The two main conclusions remain the same: 1) SignRFF is always better than SQ-RFF; 2) Non-linear LSH (SignRFF) is better than linear LSH in high similarity regime.
>
> A further explanation on the covariance term of KLSH. Because it does not have an analytical expression, this covariance term for KLSH is typically intractable and numerical simulation is also difficult because the implementation of KLSH is highly data dependent. This is why we decided to use the simplified RE if we hope to include the comparisons with KLSH.
>
> Since two reviewers are curious about the covariance term and the full RE, we might consider placing the full RE and Figures 9,10 (which do not have KLSH) in the main paper and the current Figure 1 (which has KLSH) and simplified RE in the Appendix instead.
>
> 2. The primary topic of our work is to improve the SQ-RFF method and systematically compare different LSH family algorithms by RE, as extensively experimented in Section 5. The main purpose of including deep hashing results is to show that deep hashing can be better than classical data-dependent hashing methods, but still suffer severely with moderately long codes (e.g., when $b>64$). Therefore, we provide this additional comparison with one of the latest unsupervised deep hashing method CIB [34] (published in 2021) in the paper. In [34], the authors have shown the advantage of CIB over many more deep hashing methods. Thus, we hope that our results in Section 5.1 might be sufficient. If it is believed that CIB [34] did not sufficiently compare their method with others, please kindly advise and suggest a better baseline than [34].
>
> 3. Our evaluation approach of Hamming search follows the most related (and directly comparable) SQ-RFF paper [35], as well as the long lists of papers as cited in the submission in information retrieval literature (e.g., [11,19,20,45,6,9,12,25,28,30,34,47]), and more references therein. It appears that Hamming search is a very commonly used measure for comparisons. To a large extent, this gives more direct illustration on the quality of the generated hash codes. We appreciate your comment on this and will look other ways to further strengthen the evaluations.
>
> Thanks again for your valuable feedback.

---

> > ### Author Response · Authors · 2022-08-08
> > **Feedback is welcome**
> >
> > Dear Reviewer
> >
> > Again, thank you for the constructive review comments. Please let us know if our rebuttal has adequately addressed your concerns.
> >
> > Thank you
> >
> > Authors

---

### Official Review · Reviewer_vbLr · 2022-07-13

**Rating:** 5
**Confidence:** 3
**Soundness:** 3 good
**Presentation:** 3 good
**Contribution:** 2 fair

**Summary:**

The authors propose an RFF-based binary coding method that drops the additional perturbation term from the previously popular stochastic quantization RFF method. They not only demonstrate improved performance on retrieval tasks but also prove locality-sensitivity for their proposed method.

**Questions:**

In all the cases, the KLSH performs better than SQ-RFF but never outperforms Sign-RFF. Could you elaborate more on why that is happening? What exactly is making SQ-RFF lag behind KLSH, and how sign-RFF is able to overcome that?

**Strengths And Weaknesses:**

The proposed algorithm is simple yet powerful with empirical success on the retrieval task.

---

> ### Author Response · Authors · 2022-08-02
> **Response to Reviewer vbLr**
>
> We appreciate your summary of our empirical finding that *"In all the cases, the KLSH performs better than SQ-RFF but never outperforms Sign-RFF"*,  and we hope to use this opportunity to further explain the theoretical findings by answering your question on *"Could you elaborate more on why that is happening? What exactly is making SQ-RFF lag behind KLSH, and how sign-RFF is able to overcome that?"*
>
> In the paper, we derived Equation (10), the reverse ranking probability, which motivated the definition of "Ranking Efficiency" in Definition 4 to characterize/compare the performance of different algorithms theoretically. Furthermore, Section 5.2 *Ranking Efficiency is a Predictive Measure of Search Accuracy*, provides additional explanations.
>
> For similarity search under LSH framework, we typically care more about the ranking of distances. The main motivation of our work, is to characterize the "reverse ranking probability" using Equation (10), which is the probability of outputting a wrong pairwise similarity ranking (please refer to Section 4.1 for detailed derivations). Based on (10), we developed the "ranking efficiency (RE)" in Definition 4 $\frac{E-E_c}{\sqrt{V+V_c-2V_{,c}}}$, where $E$ and $V$ correspond to specified $\rho$, $E_c$ and $V_c$ correspond to $c\rho$, for the LSH parameters $(\rho, c)$. This provides a theoretical measure to compare different LSH methods. The empirical observations you mentioned can be explained by (and all align with) the comparisons of RE in Figure 1, where we see that the RE of SignRFF is always higher than SQ-RFF, and also higher than KLSH in most cases.

---

### Official Review · Reviewer_U4Yu · 2022-07-15

**Rating:** 5
**Confidence:** 4
**Soundness:** 3 good
**Presentation:** 4 excellent
**Contribution:** 3 good

**Summary:**

The paper investigated the quality of LSH binary coding for efficient nearest neighbor retrieval. They proposed a simple strategy to extract binary codes from Random Fourier Feature (RFF) and designed a new measure called ranking efficiency to evaluate the binary coding performance of different LSH methods. Extensive experiments validated the effectiveness and consistency of ranking efficiency from theoretical evaluation and the superiority of SignRFF with sufficient binary codes.


**Questions:**

Q1. Followed by W1, as they have the formulas of collision probability for SQ-RFF and Sign-RFF (Equations 6 and 8), can they provide more empirical results (or theoretical analysis) about the variance reduction?

Q2. Followed by W2, can they follow the same settings of Figure 1 and plot the curves of $V_{xy}$ compared with $(V_x + V_y) / 2$?


**Ethics Review Area:**

["I don’t know"]

**Limitations:**

Yes.

**Strengths And Weaknesses:**

Strengths

S1. The authors simplify the binary coding based on RFF, and they demonstrate the new SignRFF is still locality-sensitive.

S2. The new measure of ranking efficiency is very interesting to me, especially its consistency in both theory and practice. As the LSH efficiency (called by the authors) mainly targets the efficiency and space overhead of LSH schemes, it seems to be a good complement measure of accuracy.

S3. The experimental results are comprehensive and convincing.

S4. This paper is well written and easy to follow.

Weaknesses

W1.I have a minor concern about the contributions of this paper, especially on the part of SignRFF. As the demonstration of the locality-sensitive property of SignRFF primarily comes from Lemma 1 [26], this part might be more solid if they could make an empirical analysis (or theoretical analysis if possible, will be better) of the variance reduction by removing the noise $\xi$.

W2. Even though the ranking efficiency (RE) is very interesting to me, the operation to directly removing the covariance $V_{xy}$ is less promising. It will be more convincing to me if they could conduct experiments to validate that the values of $V_{xy}$ for most cases of $\rho$ and $c$ are much small than the values $(V_x + V_y) / 2$.

---

> ### Author Response · Authors · 2022-08-02
> **Response to Reviewer U4Yu**
>
> The authors appreciate the reviewer's positive feedback on our idea, experiments and presentation and thank you for your questions (Q1, Q2 and related W1, W2), which would allow us to more insights to further clarify the contributions of the paper.
>
>
> Q1 (W1). Thanks for the nice suggestion to add the variance comparison (i.e., the new Figure 11, Appendix B.1, in the revision) which indeed would provide additional insight on the introduction of "ranking efficiency", because neither the collision probability (denoted by $E$) nor the variance  (i.e., $V=E(1-E)$) alone would correctly characterize the ranking efficiency.
>
> While the expressions of collision probabilities ($E$) and variances ($V=E(1-E)$) are complicated, they can be plotted for clear comparisons. In Figure 11 (left), we plot the theoretical collision probabilities of SignRFF and SQ-RFF (and other methods), for the full ranges of $\gamma$ and $\rho$. In the right plot of Figure 11, we plot the theoretical variances of SignRFF and SQ-RFF (and others).
>
> In Equation (10), our derivation has shown that the ranking efficiency is proportional to  $\frac{E-E_c}{\sqrt{V+V_c-2V_{,c}}}$, where $E$ and $V$ correspond to specified $\rho$, $E_c$ and $V_c$ correspond to $c\rho$, for the LSH parameters $(\rho, c)$. Note that the covariance term $V_{,c}$ could not be easily calculated by KLSH method. This has motivated us to define "ranking efficiency" $RE = \frac{E-E_c}{\sqrt{V+V_c}}$ in order to compare all four methods including KLSH. Figure 1 is able to show the advantage of SignRFF over SQ-RFF, in this definition of RE.
>
>
> Q2 (W2). Indeed, the covariance $V_{,c}$ does not need to be very small, although it does not affect the comparison results. Following the above explanation and your suggestion, we also define and plot the "Full RE" $\frac{E-E_c}{\sqrt{V+V_c-2V_{,c}}}$ in Figure 9 and Figure 10 in the revision. We can see that the relative comparisons between SignRFF and SQ-RFF remain similar to Figure 1.
>
> When writing the paper, We initially used the "full RE" by including the covariance term, but because this term could not be easily calculated for KLSH, we eventually decided to use current definition of RE. But Reviewer's question reminded us that it might be a good idea to add the "full RE" back at least in the Appendix. Thank you.
>
>
> A further explanation on the covariance term of KLSH. Because it does not have an analytical expression, this covariance term for KLSH is typically  intractable and numerical simulation is also difficult because the implementation of KLSH is highly data dependent. This is why we decided to use the simplified RE if we hope to include the comparisons with KLSH.
>
>
> Again, the authors thank the reviewer for the two questions and we are happy to provide answers to them which should provide additional insight to help readers further understand our work.

---

> > ### Comment · Reviewer_U4Yu · 2022-08-06
> > **Thank you for your response**
> >
> > Thank you for the new empirical results and the detailed feedback.
> >
> > The responses have addressed my concerns!

---

> > > ### Author Response · Authors · 2022-08-07
> > > **Thank you for your reply and support**
> > >
> > > Dear Referee #U4Yu,
> > >
> > > Thank you.
> > >
> > > We are very pleased to know that our response has addressed your concerns. We appreciate your thorough and supportive reviews. Please let us know if there is anything else we should try to address to help you (and all Reviewers) finalize the ratings for our submission.
> > >
> > > Sincerely,
> > >
> > > The authors

---

### Meta-Review · Area_Chair_nvho · 2022-08-26

**Recommendation:** Accept
**Confidence:** Less certain

**Metareview:**

This was a borderline paper, with three (mostly) positive reviews and one somewhat negative review.

Some of the concerns raised by the reviewers include marginal contributions as well as some issues with the empirical results (though some reviewers felt the results were strong).  It does seem that at least the first reviewer was satisfied by the rebuttal response, and the concerns of the more negative reviewer seemed to be addressed at least somewhat in the rebuttal.

I also took a close look at the paper.  I am somewhat on the fence with this paper but I think the merits outweigh any flaws and that the paper could be accepted at the conference.  The paper has two somewhat disjoint contributions (one on the SignRFF method and the other introducing the ranking efficiency measure), each of which alone is not enough to carry a NeurIPS paper.  I think together it's OK but a bit non-traditional in its presentation.  It might be good in the final version to consider (if possible) adding additional comparisons, particularly to deep hashing methods.

**Award:**

No

---

### Decision · Program_Chairs · 2022-09-14

Accept